psychology/cognition

social learning, selective trust, trait ascription, social cognition, retrospective inferences

**Author for correspondence:**
Friederike Schütte
e-mail: friederike.schuette@uni-goettingen.de

# Retrospective inferences in selective trust

Friederike Schütte[1], Nivedita Mani[2,3] and Tanya Behne[1,3]

[1]Developmental Psychology, University of Göttingen, Waldweg 26, 37073 Göttingen, Germany
[2]Psychology of Language, University of Göttingen, Goßlerstraße 14, 37073 Göttingen, Germany
[3]Leibniz ScienceCampus 'Primate Cognition', Göttingen, Germany

 FS, 0000-0002-1771-8399

Young children learn selectively from others based on the speakers' prior accuracy. This indicates that they recognize the models' (in)competence and use it to predict who will provide the most accurate and useful information in the future. Here, we investigated whether 5-year-old children are also able to use speaker reliability retrospectively, once they have more information regarding their competence. They first experienced two previously unknown speakers who provided conflicting information about the referent of a novel label, with each speaker using the same novel label to refer exclusively to a different novel object. Following this, children learned about the speakers' differing labelling accuracy. Subsequently, children selectively endorsed the object–label link initially provided by the speaker who turned out to be reliable significantly above chance. Crucially, more than half of these children justified their object selection with reference to speaker reliability, indicating the ability to explicitly reason about their selective trust in others based on the informants' individual competences. Findings further corroborate the notion that young children are able to use advanced, metacognitive strategies (trait reasoning) to learn selectively. By contrast, since learning *preceded* reliability exposure and gaze data showed no preferential looking toward the more reliable speaker, findings cannot be accounted for by attentional bias accounts of selective social learning.

## 1. Introduction

Most of what young children acquire they learn from interaction with, testimony from or observation of others [1]. This reliance on social agents for information leaves young learners vulnerable to error because others might deceive them or unintentionally provide unconventional information [2]. A plethora of recent findings has demonstrated that children are not as easily

deceived as assumed but are quite selective in whom and what they trust. When confronted with two possible sources of information, children from 4 years on show robust capacities to accept new information selectively from, for example, the more knowledgeable, older, stronger, more familiar or nicer informant (for reviews, e.g. [3,4]).

Preschool-aged children are also fairly sensitive to the informant's proven competence and trust more in the information provided by previously accurate informants than by previously inaccurate informants [5–11]. In the classic selective trust paradigm [9], children first experience two informants label familiar objects. One of the informants is consistently accurate in labelling the objects while the other is consistently inaccurate. Subsequently, children preferentially accept new information from the previously accurate, and thus reliable informant. Preschool-aged children thus predict future informativeness of sources on the basis of their previous accuracy. But what is the underlying cognitive mechanism of such selective trust?

One proposal is that children make these inferences on the basis of conceptual knowledge [12] including trait reasoning [13]. By the age of three, after listening to testimony from a consistently reliable and a consistently unreliable speaker, children can indeed identify the actor who was previously reliable [8]. Further, only those children who successfully identify the reliability of the informants actually learn selectively from the more reliable one [9]. More importantly, preschool-aged children selectively trust the model with the relevant competence for the tasks at hand [6,14]. For example, after they experience two models differing in competence in two domains, e.g. labelling accuracy and physical strength, children select models in accordance with their attributes, preferring the strong model for strength-related tasks and the knowledgeable model for knowledge-related tasks [6]. Again, this pattern of selective model choice held only for those children who correctly identified the attributes of both models indicating that children's rational choice between two models competent in different domains depends on their knowledge about the underlying traits.

By contrast, attentional bias accounts (such as the 'associative account' by Heyes [15,16]) argue that children's selective trust is not necessarily based on higher socio-cognitive processes, but can be based on low-level, attentional differences alone. When children learn about their informants' reliability, Heyes argues, they develop an attentional bias toward the more reliable speaker since her output matches children's visual input whereas the unreliable speaker's utterances constantly cause prediction errors. This bias then leads children to be more attentive to the information provided by the reliable speaker. In the classical selective trust paradigm, children thus develop an attentional bias toward the speaker labelling objects accurately and are thus more likely to acquire the novel object–label link she provides.

To test if young children are indeed able to learn selectively on the basis of trait reasoning and not (only) due to attentional differences during learning, we investigate children's selective trust in a retrospective task in which the novel information provided by two unknown informants *precedes* accuracy exposure. Thus, at the time when speakers provide novel information, children are still ignorant about the speakers' (un)reliability and a potential attentional bias developed during accuracy exposure cannot impact the selective endorsement of novel information. If children selectively endorse the information that had been provided by the speaker who subsequently turned out to be reliable, children's trust cannot be easily explained by such low-level attentional biases. Instead, it would indicate that children are able to make (metacognitive) inferences on the basis of informants' competence (trait reasoning) and, thus, to strategically learn from one source over another.

Previous research indicates that by 4–5 years of age children are able to make retrospective inferences in general [17] as well as engage in belief revision [18,19]. In fact, one previous study by Scofield & Behrend [20] suggests that some children aged 4 may revise their trust when informants proved unreliable after they had taught a new label to children. In this study, an unknown speaker used a novel label to refer to one of two novel objects, and at test, all 4-year-olds selected this object as the referent of the novel label. Subsequently, this first speaker proved to be unreliable (labelling familiar objects incorrectly) and a second speaker then appeared who used the same novel label to refer to the other novel object (as well as labelling familiar object correctly). When children were now again asked for the referent of the novel label, half of the 4-year-olds changed their original object–label link and chose the object that the second speaker had referred to, while the other half stuck with the initial object–label link provided by the unreliable source. One possible interpretation of this pattern of responses is that at least some of the 4-year-olds revised their judgement retrospectively in accordance with the source's respective reliability. Another possibility is that, once both novel objects had been presented as potential referents (given that each had now been referred to with the novel label), children's performance as a group reflects chance level, with half the children choosing one of the two novel objects and the other half chosen the other novel object. It is, thus, an open question whether

children aged 4–5 years are indeed able to revise their trust retrospectively (but see [21], for very recent positive evidence using a somewhat different paradigm).

When children encounter conflicting novel information provided by two unfamiliar speakers, will they use subsequent information about the speakers' respective reliability to evaluate the speakers' earlier claims and selectively endorse the novel information by the speaker who turned out to be the more reliable source? In the present study, 5-year-old children first experienced two unknown speakers who provided conflicting information about the referent of a novel label, with each speaker using the same novel label to refer exclusively to a different novel object. Only then did children learn about the speakers' labelling accuracy (i.e. that one of them labels familiar objects correctly, the other incorrectly).

Following the assumption that children selectively trust on the basis of trait reasoning, we predicted that they would be able to retrospectively infer that the speaker who turned out to be reliable initially provided the correct object–label link. Thus, when subsequently asked to select the correct referent for the novel label, children should be more likely to select the object initially labelled by the speaker who later turned out to be reliable than by the one who turned out to be unreliable.

We expect this effect both in a test in which objects are displayed together with the speakers, indicating selective endorsement of the object–label link provided by the reliable speaker and in a subsequent test when the objects are displayed without the speakers being present, indicating source-independent selective endorsement of the object–label link provided by the reliable speaker.

Crucially, we assume that children can only use the reliability of a speaker to make (retrospective) inferences if they also ascribe (un)reliability to each speaker in the first place. Therefore, only those children who were able to identify the speakers' reliability should selectively accept the object labelled by the reliable speaker as the referent of the novel label in the retrospective object selection tasks (both when speakers are present and absent). We do not expect such retrospective endorsement in children who were not able to correctly identify the speaker's reliability. This would also be in line with previous findings by Koenig & Harris [9] as well as Hermes *et al.* [6].

After the retrospective object selection task (with both speakers present), children were asked to justify their object decision. Following the trait-based explanation of selective learning, we expect those children who refer to speaker reliability to be more likely to endorse the information by the reliable than by the unreliable speaker in this retrospective task. In addition, we appended the 'classic selective trust paradigm' (prospective task), in which children experienced the speakers' labelling accuracy *before* encountering the conflicting novel information the two speakers provided. Replicating the classical selective trust finding (e.g. [9]), children in this prospective task should also be more likely to endorse the information provided by the previously reliable than the previously unreliable speaker, but again, only if they are able to identify the speakers' reliability.

Finally, we hypothesize that making retrospective inferences (in the context of selective trust) is more demanding than predicting future (in)accurate behaviour on the basis of past reliability (see [20]). Thus, we predict that more children will pass the prospective than the retrospective object selection task. Further, children who pass the retrospective task should also pass the prospective task (i.e. classical selective trust paradigm) but children who do not pass the prospective task should tend to not pass the more demanding retrospective task. All hypotheses, procedural details, statistical tests, sample size and exclusion criteria were preregistered on OSF (https://osf.io/bqhrx/). Both the data and analysis code can be accessed via this link as well.

# 2. Material and methods

## 2.1. Participants

Forty-two[1] 5-year-old children ($M_{age} = 65.21$ months; range: 59–71 months; 18 girls) were recruited from a database of parents who had volunteered to participate in studies on child development and came from mixed socio-economic backgrounds. Parents gave their written consent for the participation of their children. Ten additional children were tested but excluded from the analysis due to experimenter error ($n = 1$), technical error ($n = 2$) or failure to correctly answer the control questions ($n = 7$). These exclusion criteria had previously been specified in our preregistered analysis plan. All children were tested at their daycare centre in a quiet room.

---

[1]Required sample size based on a power analysis (with G\*Power). Our goal was to obtain 0.95 power to detect a small effect size of 0.25 at the standard 0.05 alpha error probability with an exact binomial one-sided test.

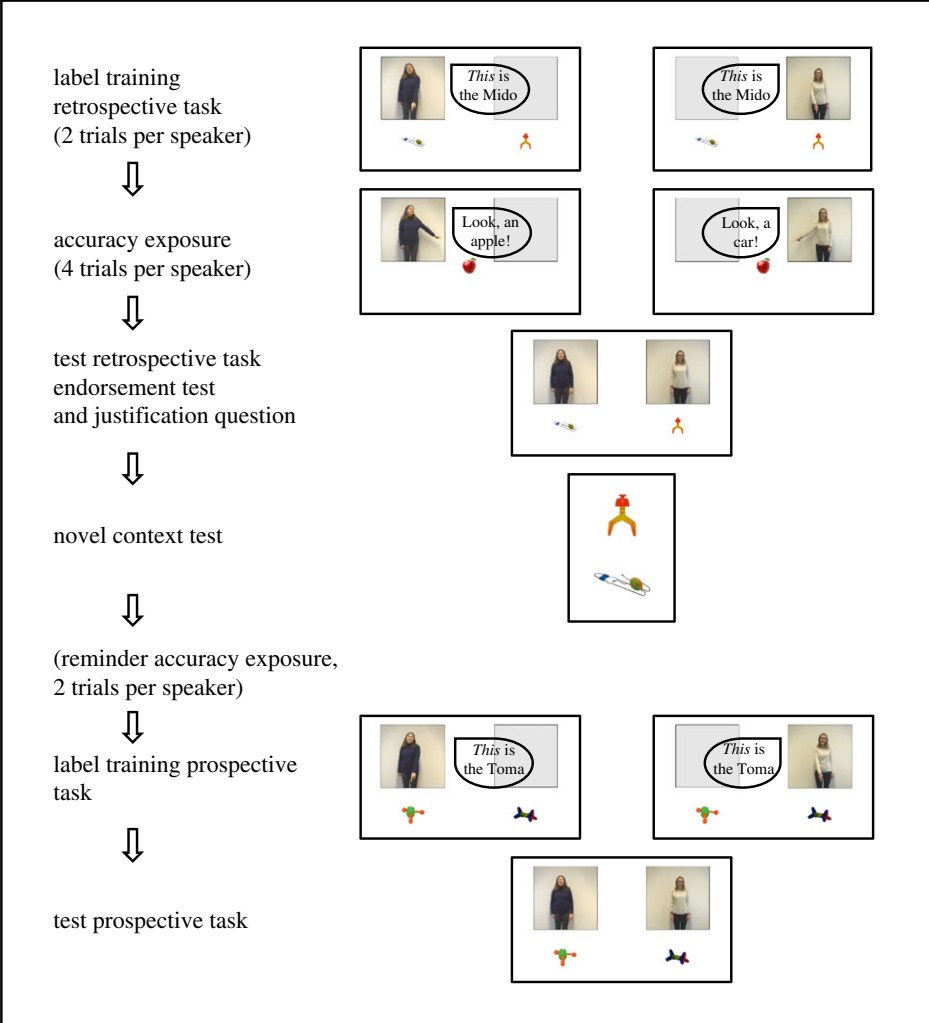

**Figure 1.** Procedural steps (left) with corresponding original screens (right). See main text for details of speakers' and experimenter's statements or question(s) in each step.

## 2.2. Materials and design

Two novel two-syllable words, 'Mido' and 'Toma' were used for label training. Four novel objects were retrieved from the *Novel Object and Unusual Name (NOUN) Database* [22] based on their low familiarity and nameability scores (figure 1). During the accuracy exposure, children saw prototypical photographs of six familiar objects.

For the introduction, label trainings and accuracy exposure trials, video clips were prepared displaying two female actors on the left and right side of the screen (figure 1). The lengths of each of the videos from the two speakers in each of the trials were matched. Further, the order of familiar objects named by the actors in the accuracy exposure trials was held constant across participants (baby, car, ball, bike, apple, dog).

Reliability of the speaker was counterbalanced across children regarding the following three aspects: The order in which the speakers appeared in label training and accuracy exposure (reliable first versus unreliable first), the actor assigned to a speaker role (speaker A versus speaker B) and the object assigned to a speaker role (orange and colourful versus blue and green).

## 2.3. Procedure

### 2.3.1. Object familiarization

Children were first familiarized with both objects through a 'game' in which they had to catch each novel object on a screen. The experimenter then explained that she builds toys and, according to her manual, she now needed a part of a toy that is labelled 'Mido' and one of the objects they had just seen was such a

'Mido', but that the experimenter did not know which one. She asked the child to help her find out which object was the Mido.

### 2.3.2. Label training retrospective task

Children saw both speakers introducing themselves one after the other in either the left or right upper corner of the screen, by saying 'hello' and waving their hands. Then, the two novel objects appeared on the left and right side of the screen underneath each of the speakers. Speaker 1 claimed 'There is *one* "Mido" here' (stressing her point by holding up her thumb to indicate 'one'), to emphasize that the objects could not both be 'Midos'. She gazed at both the objects and then pointed towards the one underneath her saying '*This* is the "Mido"' (figure 1). Speaker 2 repeated the same procedure using the same words and gestures, but claimed that the object underneath *her* was the 'Mido'. Afterwards, each speaker, in turn, looked at both objects again and claimed again '*This* is the "Mido"', pointing to the object presented underneath each of them, respectively. The alternating naming was aimed to avoid order confounds and to establish a 'balanced conflict' in which neither speaker had an immediate advantage. In order for the children to potentially select the object labelled by the speaker who was later revealed as more reliable, it was crucial that children remembered which speaker had labelled which object.

In a subsequent first episodic check, children were tested on their learned speaker-object association and prompted to point towards the respective object when asked 'Which one did she [Speaker 1/2] say is the "Mido"?'. Only if children were able to answer this question correctly for both speakers, did the experimenter proceed with the accuracy exposure. If children failed at least in one of the two questions, the second part of the label training was repeated in which both speakers labelled each of their objects as 'Mido', after which children were given the same episodic check again. Seven children failed at the first episodic check and had to re-watch the label training. Three children did not pass this control question even after repetition of the label training and were thus excluded from the study.

### 2.3.3. Accuracy exposure

The experimenter left the child alone at the PC, pretending to need to organize something. Thus, the child was on her quest by herself and could not ask the experimenter for help and was considered the 'expert' later on, being the only person to have watched the video. Each speaker labelled the same four familiar items, but only one speaker consistently labelled them accurately. The other labelled the objects consistently inaccurately (calling the baby a 'car', the apple a 'house', the ball a 'dog' and the bicycle a 'shoe'). Both speakers labelled each object with the introductory phrase 'Oh, look, this is a ___'. In order to reinforce traits of specific speakers and the contrast between them, they labelled the objects in the following order: a block of three items labelled by speaker 1, then the same block by speaker 2, then the last object first labelled by speaker 1 then speaker 2. Thus, each object was labelled twice in a row, once per speaker (order counterbalanced between children).

### 2.3.4. Test phase retrospective task

The experimenter returned and then asked the child the test question: 'Which one is the "Mido"?' referring to the screen showing the two speakers and objects (endorsement test). If children hesitated to give an answer, the experimenter explained that she needed to know which of the two objects (pointing at both, always starting with the one on the left), the child thought was more likely to be the 'Mido'.

Children who made a choice were then asked for a reason for their decision (justification question). Subsequently, to ensure that children still remembered which speaker had provided which kind of information, children went through another, second episodic check on their object–speaker association (Which speaker labelled this object as a 'Mido' earlier on?). Two children did not pass these control questions and were, thus, excluded from the study.

Two following trait questions were aimed to test whether children were aware of the speakers' differing accuracy ('Which of the women is better at labelling things? And how did the other woman label things?'). Lastly, children were asked the test question 'Which one is the "Mido"?' again, but this time while being presented with the two objects on a sheet of paper in a different spatial arrangement (on top of each other instead of side by side, figure 1) and without any of the speakers presented, neither on the sheet nor onscreen (novel context test). This additionally tested if children had selectively acquired and retained an object–label link that they could use in a somewhat different context without any supporting cues to the original source of knowledge.

### 2.3.5. Prospective task

Next children participated in a classic selective trust task (e.g. [8]), in which the information about the speakers' respective reliability is followed (not preceded) by the two speakers providing novel information. To set up this task, the experimenter shortly explained to the child that she just realized that she also needed a second part, which was called 'Toma' but that she did not know what a 'Toma' was. Children first experienced a rerun of the accuracy exposure. Each of the two speakers labelled the same two objects (car, dog), with the reliable speaker consistently labelling the two objects accurately and the unreliable speaker consistently doing so inaccurately (car—'ball', dog—'apple'), in alternating order. The reliable and unreliable speakers were the same as in the retrospective task (i.e. the same female actors with the same role assignments).

### 2.3.6. Label training prospective task

Immediately after, two novel objects appeared underneath where the speakers were located onscreen. In succession, each speaker looked at both objects and claimed, '*This* is the "Toma"', pointing to the object presented underneath each of them, respectively.

### 2.3.7. Test phase prospective task

Another, third, episodic check tested children's object–speaker association ('Which speaker labelled this part as a "Toma"?'). Two children did not pass these control questions and were, thus, excluded from the study. The children who were able to pass these questions could proceed to the test question: 'Which one is the "Toma"?'. If children gave an answer, they were also asked for a reason for their decision.

Each session lasted approximately 10 min. A back camera recorded both children's behaviour, screen and children's object selections (i.e. pointing gestures and verbal answers). An additional front camera on top of the screen recorded children's looking behaviour.

## 3. Results

### 3.1. Retrospective task performance

Children selected the object initially labelled by the reliable speaker significantly above chance in both retrospective tests (figure 2). In the endorsement test, in which objects were shown in the presence of speakers, 69% of all children selected the object initially labelled by the speaker who subsequently proved more reliable. A similar pattern emerged in the subsequent novel context test, in which the speakers were absent: 74% of all children selected the object initially labelled by the reliable speaker. Hence, children selectively endorsed the object–label link provided by the speaker who later proved to be reliable rather than unreliable (figure 2). Children showed high consistency in their choice of an object across the two retrospective tests (linear weighted $\kappa = 0.80$). In fact, those children who selected an object in both tests always selected the same object.

### 3.2. Retrospective task and object choice justification

The majority of children who passed the retrospective tests justified their object selection with reference to the speakers' reliability as demonstrated in the accuracy exposure. Specifically, 16 out of the 29 children who passed the endorsement test responded to the open question by explicitly referring to the speakers' reliability (table 1). Those children who explicitly referred to speaker reliability in the justification question showed a different pattern of responses on the retrospective tests compared to those children who did not provide such reasons (two-tailed Fisher's exact test,[2] retrospective endorsement test: $p = 0.006$, novel context test: $p = 0.016$). As predicted, those children who gave a spontaneous justification referring to speaker reliability were significantly more likely to have endorsed the object–label link

[2]Note that we preregistered a McNemar analysis to test our hypothesis concerning the relationship between retrospective test performances and reference to speaker reliability in children's spontaneous justifications. This showed that those children who explicitly referred to speaker performance in labelling familiar objects were significantly more likely to have endorsed the label provided by the reliable rather than the unreliable speaker (one-sided McNemars, both retrospective tests, $p < 0.001$). However, as reviewer one pointed out, a more relevant analysis to test our hypotheses would be between- rather than within-group analyses. Thus, in addition to these preregistered analyses, we performed Fisher's exact test and binomial tests, as reported in the main text.

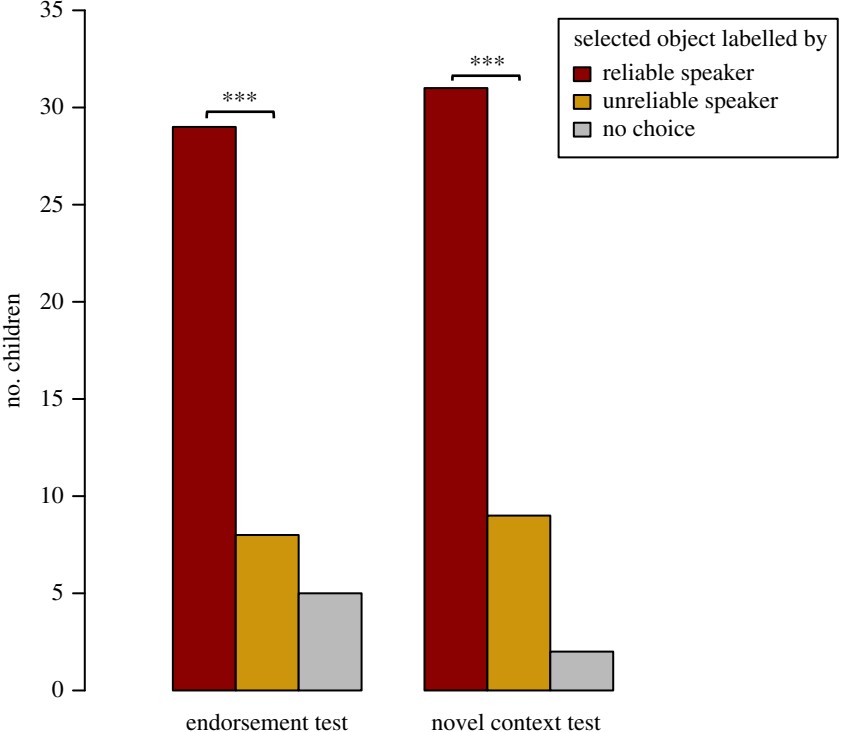

**Figure 2.** Object selection in the retrospective tests. In each retrospective test, significantly more children chose the object initially labelled by the reliable rather than the unreliable speaker. In the endorsement test, 37 children selected one of the two objects, of which 29 selected the object initially labelled by the reliable speaker ($p < 0.001$ one-sided binomial (see footnote 3)). In addition, two children selected both objects and three selected none. In the novel context test, 40 children made an object choice, of which 31 selected the object initially labelled by the reliable speaker ($p < 0.001$, one-sided binomial). Further, one child selected both, and one child selected none.

**Table 1.** Overview of children's object choice justifications (with versus without reference to the speakers' reliability) divided into test performances in both retrospective tests and the prospective test. Note that the one child that failed the retrospective tests but still justified her decision with reference to the speakers' reliability is the same child for both endorsement and novel context test.

|  |  | explicit reference to speaker reliability | |
|---|---|---|---|
|  |  | yes | no |
| retrospective endorsement test | pass | 16 | 13 |
|  | fail | 1 | 12 |
| retrospective novel context test | pass | 16 | 15 |
|  | fail | 1 | 10 |
| prospective test | pass | 12 | 18 |
|  | fail | 0 | 12 |

provided by the reliable rather than by the unreliable speaker (one-sided exact binomial test, $p < 0.001$ for both retrospective tests). By contrast, those children who did not make any spontaneous reference to speaker reliability were not more likely to have endorsed the object–label link provided by the reliable rather than by the unreliable speaker (one-sided exact binomial test, $p = 0.5$ for the retrospective endorsement test and $p = 0.212$ for the novel context test).

## 3.3. Gaze duration analyses

Since label-object learning preceded the accuracy exposure, children's selective endorsement cannot be explained by an attentional bias developed during accuracy exposure. However, in the retrospective

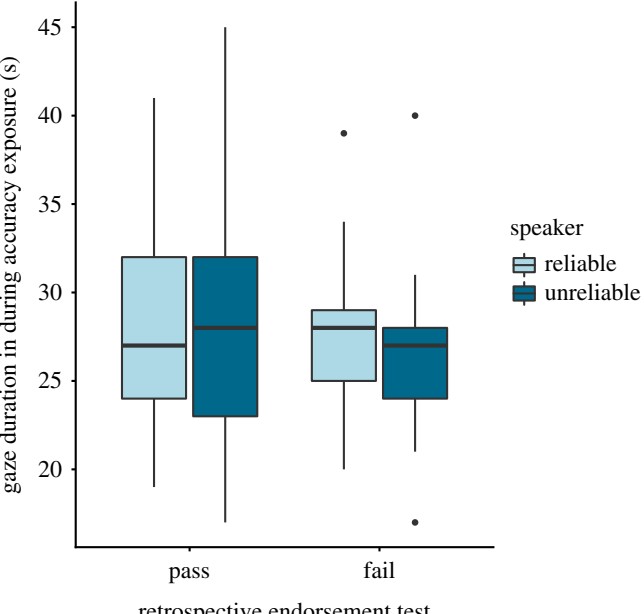

**Figure 3.** Standard boxplot displaying children's looking times at the reliable and unreliable speaker during the main accuracy exposure as a function of task performance in the retrospective endorsement test (i.e. with the speakers present). (Passing task = choosing the object initially labelled by the reliable speaker, failing = choosing the object initially labelled by the unreliable speaker, both or none of the objects). Analyses showed that children who passed the endorsement test did not look significantly longer to the reliable speaker (M = 27.96 s; s.d. = 5.92 s) than the unreliable speaker during the accuracy exposure (M = 28 s; s.d. = 6.63; p = 0.971, paired t-test). Furthermore, children who endorsed the object–label link provided by the reliable speaker did not look significantly longer to that speaker during accuracy exposure (M = 27.97 s; s.d. = 5.92 s) than children who did not endorse this link (M = 27.85 s; s.d. = 4.88 s; p = 0.946, independent t-test).

endorsement test directly following the accuracy exposure, pictures of the speakers were presented accompanied by their respective objects on the same side as in the preceding accuracy exposure (see figure 1). If children developed an attentional bias toward the reliable speaker in the accuracy exposure, this bias could potentially still transfer to the following test and lead children to select the object on the *side* of the reliable speaker, without considering who initially labelled which object. Thus, we analysed the relationship between children's gaze durations during the accuracy exposure and their performance in the retrospective endorsement test. For each speaker and accuracy exposure trial, we manually measured the duration of time children spent looking at the speaker and the object she labelled. We then calculated a sum of their looking times across these four trials. This yielded the absolute gaze duration towards each speaker during the accuracy exposure. The first analysis, which we had planned, operationalized children's gaze duration during the accuracy exposure as a binary variable (looking longer at the reliable versus the unreliable speaker). Children's object selection in the Endorsement test was not significantly related to their relative gaze duration at the speakers during accuracy exposure (p = 0.405, two-sided McNemar). Further, exploratory analyses that looked at children's gaze duration as a continuous variable showed the same pattern of results (figure 3). Thus, we could not detect any relation between children's visual attention during the accuracy exposure and their performance in the retrospective endorsement test.

## 3.4. Prospective test performance and the relation between prospective and retrospective tests

In the prospective test, children selected the object labelled by the reliable speaker significantly above chance (30 of those 39 children who made a choice, p < 0.001, one-sided binomial[3]). Twelve of

[3]Note that we also preregistered a multiple logistic regression analysis in which object choice (blue/orange) would be predicted by the object labelled by the reliable speaker (blue/orange) and performance on trait questions (pass/fail). As explained in the preregistration, this analysis should only be performed if the less frequent response (object selection: blue versus orange) occurred at least 20 times (no. of predictors × 10; [23]). Since the less frequent outcome (blue object selection) occurred only 18 times in both retrospective tests and only 16 times in the prospective test, these analyses were not performed.

**Table 2.** Children's performance in the two retrospective tests as a function of their performance in the prospective task. Note that 'fail' in all tests here includes all children who did not exclusively choose the object initially labelled by the reliable speaker, and thus, all children who chose the object initially labelled by the unreliable speaker as well as those who did not choose any object or chose both objects.

| | | prospective test | |
|---|---|---|---|
| | | pass | fail |
| retrospective endorsement test | pass | 22 | 7 |
| | fail | 8 | 5 |
| retrospective novel context test | pass | 25 | 6 |
| | fail | 5 | 6 |

these children explicitly referred to speaker performance in labelling familiar objects when asked to justify their object selection. Those children who explicitly referred to speaker reliability in the justification question showed a different pattern of responses on the prospective test compared to those children who did not provide such reasons (exploratory two-sided Fisher's exact test, $p = 0.009$). Those children who gave a spontaneous justification referring to speaker reliability were significantly more likely to have endorsed the object–label link provided by the reliable rather than by the unreliable speaker (one-sided exact binomial test, $p < 0.001$). By contrast, those children who did not make any spontaneous reference to speaker reliability were not more likely to have endorsed the object–label link provided by the reliable rather than by the unreliable speaker (one-sided exact binomial test, $p = 0.6$).

Overall, children's pattern of performance in the prospective test was similar to their performance in the retrospective tests (table 2). Contrary to our predictions, children were not more likely to fail a retrospective test but pass the prospective task than to show the reverse pattern of performance (endorsement test: $p = 0.5$, one-sided McNemar; novel context test: $p = 0.726$, one-sided McNemar).

## 3.5. Trait ascription and performance on the selective trust tasks

The trait ascription task was passed by 95% of children. In other words, all but two children answered both trait questions correctly. For each retrospective test, significantly more children were able to answer both trait questions, but still failed the retrospective test, than showed the reverse pattern of performance (i.e. were not able to answer both trait questions, but passed the retrospective test; retrospective endorsement test, $p = 0.002$; retrospective novel context test, $p = 0.006$; one-sided McNemar). The same pattern holds when looking at the relation of the trait ascription task to the prospective task ($p = 0.003$, one-sided McNemar).

## 4. Discussion

In order to learn efficiently about the world around them, it is crucial for children to develop the ability to critically evaluate their sources and, in the case of conflict or uncertainty, select some information over others. Although a plethora of previous research suggests that preschool-aged children selectively accept information from previously reliable over unreliable informants (e.g. [1]), the cognitive processes underlying this selective behaviour are still unclear. To further investigate which mechanism drives children's learning, we here used a reversed version of the classic selective learning paradigm and demonstrate that young children use speaker reliability to selectively accept the more reliable source *in retrospect*. Children selectively endorsed the object–label link initially provided by the speaker who later turned out to be reliable. This even transferred to a somewhat novel context that did not include the original sources.

These main findings using the reversed paradigm suggest that children's selective trust in the more reliable source cannot be easily explained by an attentional bias toward the reliable speaker biasing children to attend to new information s/he provides since label introduction was *prior* to accuracy exposure.

To pass the first endorsement test in which the two novel objects appeared underneath the speakers, one might still argue that it was possible for the children to merely select the object on the side of the speaker they paid more attention to during accuracy exposure. However, we obtained the same reliability effect in the subsequent, novel context test, in which objects were spatially rearranged and speakers' pictures were absent. Furthermore, additional analyses of gaze data did not find any relation between children's visual attention during the accuracy exposure and their performance in the retrospective endorsement test, further supporting the notion that their selective behaviour might not be a result of attentional processes alone.

By contrast, more than half of the children justified their object selection with reference to speaker reliability, indicating advanced metacognitive awareness and the ability to explicitly reason about their selective trust in others based on the informants' individual competences. Thus, our findings are in line with conceptual accounts of selective trust [12] including trait reasoning [6]. Children's retrospective weighting of information based on informants' reliability here are in accordance with related findings on 6.5-year-old children's learning from ignorant versus knowledgeable speakers [24]. Extending previous research, our study suggests that even younger children use trait reasoning to learn selectively from others and do this even in situations in which reliability has to be inferred (from the previous accuracy) rather than explicitly stated (proclaimed knowledge/ignorance).

A very recent study supports this claim: Luchkina et al. [21] presented children at around 5 years of age with one speaker who labelled two novel objects and then continued to label familiar objects. In a between-subject design, the speaker either turned out to be accurate (i.e. she labelled the familiar objects correctly) or inaccurate (she labelled the familiar objects incorrectly). Children relied on the novel information the speaker had provided significantly more in the accurate than the inaccurate speaker condition. Thus, using somewhat different methodologies, the research findings obtained all suggest that preschool-aged children discount information from inaccurate speakers retrospectively [20,21].

How does children's performance in the retrospective task relate to their performance on standard selective trust tasks? In our prospective task, in which children received novel information after experiencing the speakers' accuracy, they also selectively endorsed the information provided by the reliable speaker. Thus, our study additionally replicates previous findings showing that 5-year-old children selectively predict future accuracy on the basis of speakers' prior reliability (e.g. [8,25]). We assumed that this prediction is less cognitively demanding than making retrospective inferences in selective trust. The prospective task could potentially be solved by associative learning on the basis of biased attention alone. However, the number of children passing both tasks was similar and passing the retrospective task was not predicted by children's performance on the prospective task. Although the prospective task might be cognitively less demanding, this advantage could be counterbalanced by the fact that there is a stronger pressure to resolve the (initially presented) conflict in the retrospective task, leading more children to use reliability as a first useful cue to resolve this conflict between competing referents for the same label. This might enable even younger children to deploy informant reliability to select useful information over others, which is for future research to follow up on. Similarly, further studies might disentangle if children use speaker reliability only if available (as in the present study) or actively search for informative clues in such situations of conflict.

Young learners' ability to be selective is crucial for social and, more generally, cultural learning [1,4]. Being able to evaluate sources and selectively endorse information not only prospectively but also retrospectively, allows for greater flexibility and ecological validity of selective trust. The present study thus further suggests that preschool-aged children are already well equipped to use the characteristics of social agents to efficiently learn about the world around them.

Ethics. Ethics approval was granted by the ethics committee of the Georg-Elias-Müller-Institute for Psychology, the University of Göttingen (project no. 213). Parents signed a written informed consent form for their child.

Data accessibility. Data and analysis code are available on the Open Science Framework (OSF) under https://osf.io/bqhrx/.

Authors' contributions. T.B. and N.M. had the original research idea and acquired funding. F.S., N.M. and T.B. designed the study. F.S. collected and analysed the data with T.B.'s support. F.S. and T.B. wrote the manuscript. F.S., N.M. and T.B. discussed the results and commented on the final version of the manuscript.

Competing interests. We declare we have no competing interests.

Funding. This study was funded by Deutsche Forschungsgemeinschaft (German Research Foundation) (RTG 2070 Understanding Social Relationships) and Leibniz ScienceCampus 'Primate Cognition'.

Acknowledgements. We thank the families and their children for participating in the study. Further, we would like to thank Marlen Kaufmann, Marlene Meyer, Joana Lonquich, Luise Raussendorf and Konstanze Schirmer for their help with recruitment and coding.

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
