## [Reviewer comments · Royal Society Open Science]

Review History

RSOS-191451.R0 (Original submission)

Review form: Reviewer 1

Is the manuscript scientifically sound in its present form?

Yes

Are the interpretations and conclusions justified by the results?

Yes

Is the language acceptable?

Yes

Do you have any ethical concerns with this paper?

No

Have you any concerns about statistical analyses in this paper?

Yes

Recommendation?

Accept with minor revision (please list in comments)

Comments to the Author(s)

The manuscript reports on a study investigating the underlying mechanisms of children's selective learning from different social sources. It approaches the question by outlining two opposing theoretical perspectives; one proposing children's selective learning is a consequence of an attentional bias induced by prediction errors in the case of unreliable sources; and the other proposing that selective trust is based on more meta-cognitive trait-ascription to the sources. The paper is well written, reviews relevant literature, and reports on a neatly designed study, the results of which support the trait-ascription mechanism of selective learning. I think the manuscript is appropriate for publication in Royal Society Open Science, but outline some minor comments / suggestions for improvement below.

Introduction

- p.3, paragraph 1-2: Somewhat confusing why you seemingly contrast 'knowledgeability' and 'competence/reliability' of informants.
- p.4, paragraph 2: Couldn't one make the opposite prediction in the attentional bias account of selective learning - that prediction errors in case of mislabeling could also lead to the opposite effect; i.e. increased attention due to the surprising information?

Methods:

- Please report the age (also) in years. Beyond 24 months, I think reporting age in months is uninformative.
- It is not clear what were the labels used by the unreliable speaker. Were the labels just shuffled around? Other known words? Made-up words? Please clarify also how the labelling was done - were the labels imbedded into sentences, how many times was each familiar object labeled?

Results

- While in the tables and (most) graphs you group the children into ones that 'passed' and ones that 'failed'; you report the statistics in text for different total Ns in each test - i.e. children that had endorsed Reliable out of those who made a choice (presumably excluding those who chose both or none). It would be easier to interpret and follow the results if all percentages and all tests were done consistently - reporting the proportion of children out of the whole sample.
- Justification - you report that the children who explicitly referred to the speakers' reliability were more likely to have endorsed the reliable than the unreliable object-label link. What about the comparison between children that did and children that did not refer to the speakers' reliability? Was it not your hypothesis that those that will explicitly justify their choice will be the ones more likely making the correct choice? To speak to this hypothesis, the two groups of children should be compared instead of (or in addition to) the within-subject tests. This applies to both retro- and prospective tasks.
- I like the control for the attention during the accuracy exposure, but feel that the way this was established is not described in enough detail. Given that the speakers were not on screen at the same time - was the criteria for looking more at reliable/unreliable established by measuring looking on/off screen during each trial and then comparing the sums? Since gaze pattern was not in fact measured (if I understand correctly), I would avoid this term, as it implies more detailed analysis (where on screen they looked etc).
- I doubt that this is in fact an issue - because of your counterbalancing - but when reading your control measure I wondered why you did not report the same attention measures for the Label training phase of the experiment. It is in principle possible that children happened to attend to the speakers that turned out to be unreliable more in the training phase and the following accuracy exposure was irrelevant. I appreciate the chances for this to explain the results are minimal, but it would be good if this measure was included for the sake of any reader that, like me, might wonder if this was controlled for.
- Trait ascription - the report on the results here is confusing, please elaborate on what exactly was compared ('the reverse pattern' is ambiguous as to what you are referring to).

General terminology: "preschooler" is a relative term (5-year-olds are school-aged children in some countries) - please use a more specific term; "object-label-link" and "object-label link" and

"label-object-link" and "referent-label link" are used interchangeably - please be consistent (I would suggest object-label link).

Review form: Reviewer 2 (Emily Mather)

Is the manuscript scientifically sound in its present form?

Yes

Are the interpretations and conclusions justified by the results?

Yes

Is the language acceptable?

Yes

Do you have any ethical concerns with this paper?

No

Have you any concerns about statistical analyses in this paper?

No

Recommendation?

Accept with minor revision (please list in comments)

Comments to the Author(s)

This study investigates 5-year-olds' ability to selectively learn novel labels from adult speakers based on speaker reliability. The study pits different theoretical accounts (low-level attention vs. conceptual) against each other by requiring retrospective inference of speaker reliability in contrast to the typical prospective inference task. That is, the speakers label novel objects prior to either accurately or inaccurately labelling familiar objects. Following this accuracy exposure, children had to decide who used the novel label correctly. In an endorsement test (where speakers were present) most children selected the novel object previously labelled by the reliable speaker. A similar result was found the following novel context test (where speakers were absent). There was high consistency across children in their choices across these two retrospective tests. The results are interpreted as evidence in favour of a conceptual account rather than as the outcome of an attentional bias towards the more reliable speaker.

The manuscript is well-written and addresses an important theoretical question. I have a few comments I would like to see addressed:

1. It was not clear from the introduction whether there is any empirical evidence in favour of attentional bias accounts. Do these accounts propose overt visual attention to the more reliable speaker, or could it take the form of covert attention?
2. I couldn't find any mention of how children's looking behaviour was measured. I think this should be included, given the importance for the interpretation of children's retrospective task performance.
3. Were the speakers in the prospective task different to those in the retrospective task?
4. As the authors note, an attentional bias towards the more reliable speaker could lead children during the endorsement test to choose the object on the same side as that speaker. The authors suggest that the similar performance during the novel context test provides evidence against this

possibility. However, performance during the novel context test could be driven by the formation of an object-label mapping during the endorsement test, which in turn could be based on an attentional bias.

5. The authors further argue that their looking time data provides evidence against an attentional bias account, as there was no relation between attention during the accuracy exposure and performance in the endorsement test. This again prompts my question about whether the attentional bias accounts specifically predict overt attention to the more reliable speaker? Possibly, a more robust test would be to remove the endorsement test and move straight to the novel context test – which cannot be contaminated by attention to the reliable speaker.

MINOR

The authors argue that their findings with ‘even younger children’ are consistent with the literature on trait reasoning in children. It would be helpful to know the ages of children in these previous studies.

Why is the Luchkina et al (2018) study not mentioned in the introduction? It appears relevant to motivating the present study.

Decision letter (RSOS-191451.R0)

02-Jan-2020

Dear Ms Schütte

On behalf of the Editors, I am pleased to inform you that your Manuscript RSOS-191451 entitled "Retrospective Inferences in Selective Trust" has been accepted for publication in Royal Society Open Science subject to minor revision in accordance with the referee suggestions. Please find the referees' comments at the end of this email.

The reviewers and handling editors have recommended publication, but also suggest some minor revisions to your manuscript. Therefore, I invite you to respond to the comments and revise your manuscript.

- Ethics statement

- Data accessibility

If you wish to submit your supporting data or code to Dryad (<http://datadryad.org/>), or modify your current submission to dryad, please use the following link:
<http://datadryad.org/submit?journalID=RSOS&manu=RSOS-191451>

- **Competing interests**

- **Authors' contributions**

- **Acknowledgements**

- **Funding statement**

Because the schedule for publication is very tight, it is a condition of publication that you submit the revised version of your manuscript before 11-Jan-2020. Please note that the revision deadline will expire at 00.00am on this date. If you do not think you will be able to meet this date please let me know immediately.

If your manuscript is newly submitted and subsequently accepted for publication, you will be asked to pay the article processing charge, unless you request a waiver and this is approved by Royal Society Publishing. You can find out more about the charges at <https://royalsocietypublishing.org/rsos/charges>. Should you have any queries, please contact openscience@royalsociety.org.

on behalf of Dr Teodora Gliga (Associate Editor) and Essi Viding (Subject Editor)
openscience@royalsociety.org

Reviewer comments to Author:

Reviewer: 1

Comments to the Author(s)

The manuscript reports on a study investigating the underlying mechanisms of children's selective learning from different social sources. It approaches the question by outlining two opposing theoretical perspectives; one proposing children's selective learning is a consequence of an attentional bias induced by prediction errors in the case of unreliable sources; and the other proposing that selective trust is based on more meta-cognitive trait-ascription to the sources. The paper is well written, reviews relevant literature, and reports on a neatly designed study, the results of which support the trait-ascription mechanism of selective learning. I think the manuscript is appropriate for publication in Royal Society Open Science, but outline some minor comments / suggestions for improvement below.

Introduction

- p.3, paragraph 1-2: Somewhat confusing why you seemingly contrast 'knowledgeability' and 'competence/reliability' of informants.

- p.4, paragraph 2: Couldn't one make the opposite prediction in the attentional bias account of selective learning - that prediction errors in case of mislabeling could also lead to the opposite effect; i.e. increased attention due to the surprising information?

Methods:

- Please report the age (also) in years. Beyond 24 months, I think reporting age in months is uninformative.

- It is not clear what were the labels used by the unreliable speaker. Were the labels just shuffled around? Other known words? Made-up words? Please clarify also how the labelling was done - were the labels imbedded into sentences, how many times was each familiar object labeled?

Results

- While in the tables and (most) graphs you group the children into ones that 'passed' and ones that 'failed'; you report the statistics in text for different total Ns in each test - i.e. children that had endorsed Reliable out of those who made a choice (presumably excluding those who chose both or none). It would be easier to interpret and follow the results if all percentages and all tests were done consistently - reporting the proportion of children out of the whole sample.

- Justification - you report that the children who explicitly referred to the speakers' reliability were more likely to have endorsed the reliable than the unreliable object-label link. What about the comparison between children that did and children that did not refer to the speakers' reliability? Was it not your hypothesis that those that will explicitly justify their choice will be the ones more likely making the correct choice? To speak to this hypothesis, the two groups of children should be compared instead of (or in addition to) the within-subject tests. This applies to both retro- and prospective tasks.

- I like the control for the attention during the accuracy exposure, but feel that the way this was established is not described in enough detail. Given that the speakers were not on screen at the same time - was the criteria for looking more at reliable/unreliable established by measuring looking on/off screen during each trial and then comparing the sums? Since gaze pattern was not in fact measured (if I understand correctly), I would avoid this term, as it implies more detailed analysis (where on screen they looked etc).

- I doubt that this is in fact an issue - because of your counterbalancing - but when reading your control measure I wondered why you did not report the same attention measures for the Label training phase of the experiment. It is in principle possible that children happened to attend to the speakers that turned out to be unreliable more in the training phase and the following accuracy exposure was irrelevant. I appreciate the chances for this to explain the results are minimal, but it would be good if this measure was included for the sake of any reader that, like me, might wonder if this was controlled for.

- Trait ascription - the report on the results here is confusing, please elaborate on what exactly was compared ('the reverse pattern' is ambiguous as to what you are referring to).

General terminology: "preschooler" is a relative term (5-year-olds are school-aged children in some countries) - please use a more specific term; "object-label-link" and "object-label link" and "label-object-link" and "referent-label link" are used interchangeably - please be consistent (I would suggest object-label link).

Reviewer: 2

Comments to the Author(s)

This study investigates 5-year-olds' ability to selectively learn novel labels from adult speakers based on speaker reliability. The study pits different theoretical accounts (low-level attention vs. conceptual) against each other by requiring retrospective inference of speaker reliability in contrast to the typical prospective inference task. That is, the speakers label novel objects prior to either accurately or inaccurately labelling familiar objects. Following this accuracy exposure, children had to decide who used the novel label correctly. In an endorsement test (where speakers were present) most children selected the novel object previously labelled by the reliable speaker. A similar result was found the following novel context test (where speakers were absent). There was high consistency across children in their choices across these two retrospective tests. The results are interpreted as evidence in favour of a conceptual account rather than as the outcome of an attentional bias towards the more reliable speaker.

The manuscript is well-written and addresses an important theoretical question. I have a few comments I would like to see addressed:

1. It was not clear from the introduction whether there is any empirical evidence in favour of attentional bias accounts. Do these accounts propose overt visual attention to the more reliable speaker, or could it take the form of covert attention?
2. I couldn't find any mention of how children's looking behaviour was measured. I think this should be included, given the importance for the interpretation of children's retrospective task performance.
3. Were the speakers in the prospective task different to those in the retrospective task?
4. As the authors note, an attentional bias towards the more reliable speaker could lead children during the endorsement test to choose the object on the same side as that speaker. The authors suggest that the similar performance during the novel context test provides evidence against this possibility. However, performance during the novel context test could be driven by the formation of an object-label mapping during the endorsement test, which in turn could be based on an attentional bias.
5. The authors further argue that their looking time data provides evidence against an attentional bias account, as there was no relation between attention during the accuracy exposure and performance in the endorsement test. This again prompts my question about whether the attentional bias accounts specifically predict overt attention to the more reliable speaker? Possibly, a more robust test would be to remove the endorsement test and move straight to the novel context test - which cannot be contaminated by attention to the reliable speaker.

MINOR

The authors argue that their findings with 'even younger children' are consistent with the literature on trait reasoning in children. It would be helpful to know the ages of children in these previous studies.

Why is the Luchkina et al (2018) study not mentioned in the introduction? It appears relevant to motivating the present study.

Author's Response to Decision Letter for (RSOS-191451.R0)

See Appendix A.

Decision letter (RSOS-191451.R1)

28-Jan-2020

Dear Ms Schütte,

It is a pleasure to accept your manuscript entitled "Retrospective inferences in selective trust" in its current form for publication in Royal Society Open Science. The comments of the reviewer(s) who reviewed your manuscript are included at the foot of this letter.

on behalf of Dr Teodora Gliga (Associate Editor) and Essi Viding (Subject Editor)
openscience@royalsociety.org

Appendix A

Thursday, 23 January 2020

Dear Mr. Dunn,

We are delighted to hear that our manuscript RSOS-191451 entitled “Retrospective inferences in selective trust” has been accepted for publication in Royal Society Open Science subject to minor revisions in accordance with the referee suggestions. We appreciate the time and effort that you and the reviewers have dedicated to providing your valuable feedback on our manuscript. We are grateful to the reviewers for their insightful comments on the paper and have been able to incorporate changes to reflect most of the suggestions provided by the reviewers. Note that we uploaded two versions of the revised manuscript: One version with tracked changes (highlighted in yellow) and one ‘clean’ version without the changes highlighted.

Here is a point-by-point response to the reviewers’ comments:

Comments from reviewer 1

- *P.3, paragraph 1-2: Somewhat confusing why you seemingly contrast 'knowledgeability' and 'competence/reliability' of informants.*

We had used reliability/unreliability to refer to previous accuracy/inaccuracy and thus had contrasted this with the speaker’s stated knowledgeability/ignorance. To avoid any confusion, we have changed this and now specifically refer to ‘previous accuracy’ throughout these sections (P. 3, paragraph 2).

- *P.4, paragraph 2: Couldn't one make the opposite prediction in the attentional bias account of selective learning - that prediction errors in case of mislabeling could also lead to the opposite effect; i.e. increased attention due to the surprising information?*

We agree that one could very well argue that higher prediction error (as produced by mislabeling) could lead to *more* attention. But Heyes (2016; 2017, *Dev. Sci.*) explains it rather in terms of “low predictiveness” (p. 10), thus gradually shifting the attention away from the informant and her behavior. Most important for our purposes is that successful retrospective selective learning in our reversed order of the paradigm should rule out *any* attentional explanation of the effect, regardless of the direction of attention.

- *Please report the age (also) in years. Beyond 24 months, I think reporting age in months is uninformative.*

We changed the age report accordingly on page 7 (last paragraph).

- *It is not clear what were the labels used by the unreliable speaker. Were the labels just shuffled around? Other known words? Made-up words? Please clarify also how the labelling was done - were the labels imbedded into sentences, how many times was each familiar object labeled?*

The alternative labels used by the inaccurate speaker were words that are familiar to children at that age, but structurally dissimilar to the corresponding accurate word used by the reliable speaker (no rhyme or overlapping first phoneme). We added the alternative labels used by the inaccurate speaker, the labeling phrase used by both speakers and the number of labelling events per object to the description of the main accuracy exposure on page 10 (first paragraph). We also added the alternative labels used by the inaccurate speaker in the reminder accuracy exposure on page 11 (paragraph 2).

- *While in the tables and (most) graphs you group the children into ones that 'passed' and ones that 'failed'; you report the statistics in text for different total Ns in each test - i.e. children that had endorsed Reliable out of those who made a choice (presumably excluding those who chose both or none). It would be easier to interpret and follow the results if all percentages and all tests were done consistently - reporting the proportion of children out of the whole sample.*

We acknowledge that the way we reported the statistics with regards to different total Ns was confusing in the submitted manuscript. We changed both the report in the main text (see p. 12, paragraph 2) as well as figure 2 and figure caption 2 to report this more consistently.

- *Justification - you report that the children who explicitly referred to the speakers' reliability were more likely to have endorsed the reliable than the unreliable object-label link. What about the comparison between children that did and children that did not refer to the speakers' reliability? Was it not your hypothesis that those that will explicitly justify their choice will be the ones more likely making the correct choice? To speak to this hypothesis, the two groups of children should be compared instead of (or in addition to) the within-subject tests. This applies to both retro- and prospective tasks.*

We appreciate the concern and realize that the analysis proposed by reviewer 1 is a more relevant analysis to test our prediction. We have thus changed our analyses for the retro- and prospective task accordingly (and report the results of the preregistered McNemar tests only in footnote 3). See changes on p. 12 (last paragraph, ending on page 13) and 14 (paragraph 2).

- *I like the control for the attention during the accuracy exposure, but feel that the way this was established is not described in enough detail. Given that the speakers were not on screen at the same time - was the criteria for looking more at reliable unreliable established by measuring looking on/off screen during each trial and then comparing the sums?*

A front camera above the screen recorded children's eye movements. Although manual coding of the areas children looked at via this recording is not as fine-grained as professional eye-tracking, it sufficed for our purposes: Based on the fixations on the different speakers in the label training, coders could easily identify looks toward a speaker, the object in the center, as well as looks away from the screen. Looks at each speaker were thus recorded manually for each trial of the accuracy exposure and summed up. For each speaker and accuracy exposure trial, we manually measured the duration of time children spent looking at the speaker and the object she labelled. We then calculated a sum of their looking times across these four trials. This yielded the absolute gaze duration towards each speaker during the accuracy exposure. Our first analysis of looking time in relation to retrospective endorsement test performance treats these absolute gaze durations as a binary variable (looking longer at the reliable vs. the unreliable speaker). We added a detailed description of the measurement (page 12, paragraph 1; p. 13, paragraph 2)

- *Since gaze pattern was not in fact measured (if I understand correctly), I would avoid this term, as it implies more detailed analysis (where on screen they looked etc).*

Thank you for pointing that out, we changed the term to "gaze duration" (p. 13, paragraph 2)

- *I doubt that this is in fact an issue - because of your counterbalancing - but when reading your control measure I wondered why you did not report the same attention measures for the Label training phase of the experiment. It is in principle possible that children happened to attend to the speakers that turned out to be unreliable more in the training phase and the following accuracy exposure was irrelevant. I appreciate the chances for this to explain the results are minimal, but it would be good if this measure was included for the sake of any reader that, like me, might wonder if this was controlled for.*

We acknowledge that there might be other factors influencing selective learning. For this reason, we counterbalanced speakers and their reliability role and speaker's reliability only became apparent *after* the label training. Any systematic attentional bias toward (and hence, potentially more learning from) the speaker who later turns out to be reliable is thus very unlikely.

- *Trait ascription - the report on the results here is confusing, please elaborate on what exactly was compared ('the reverse pattern' is ambiguous as to what you are referring to).*
“The reverse pattern” refers to the opposite discordant cell in the contingency table that the McNemar test takes into account. In order to avoid confusion, we added the phrase with explicit reference to the group in parentheses (page 15, paragraph 2).
- *General terminology: "preschooler" is a relative term (5-year-olds are school-aged children in some countries) - please use a more specific term*

Yes, that is true, of course. We have adjusted our wording accordingly, referring to children's ages specifically where required or using more general terms depending on the context.

- *"object-label-link" and "object-label link" and "label-object-link" and "referent-label link" are used interchangeably - please be consistent (I would suggest object-label link).*

We now use the term “object-label link” consistently throughout the revised manuscript.

Comments from reviewer 2

- *It was not clear from the introduction whether there is any empirical evidence in favour of attentional bias accounts. Do these accounts propose overt visual attention to the more reliable speaker, or could it take the form of covert attention?*

We are not aware of any empirical evidence in favor of this account. Heyes (2016; 2017, *Dev. Sci.*) argues that most selective trust findings *could* easily be explained by a lower-level attentional bias toward the more predictive cue/person and thus, away from the unreliable speaker. Thus, in her view, previous studies could not successfully rule out such a lower-level mechanism driving the effect. We now state explicitly that the attentional bias is Heyes' point of view (page 4, paragraph 2).

Regarding the type of attention, e.g., Heyes (2016, *Dev. Sci.*) acknowledges that the attentional shift/bias does not necessarily have to be reflected in looking time. This is one reason why we did not exclusively rely on our looking time measure to test the attentional bias account, but as an additional measure. Our main objective was to test if children could learn selectively *in retrospect*, which would make an attentional explanation of the effect unlikely in the first place.

- *I couldn't find any mention of how children's looking behaviour was measured. I think this should be included, given the importance for the interpretation of children's retrospective task performance.*

This was clearly missing in the manuscript. For a description, see both the response to a similar comment by reviewer 1 above and our additions regarding this measurement in the manuscript (page 12, paragraph 1; p. 13, paragraph 2).

- *Were the speakers in the prospective task different to those in the retrospective task?*

No, these were the same speakers. We clarified this point now on page 12 (paragraph 2).

- *As the authors note, an attentional bias towards the more reliable speaker could lead children during the endorsement test to choose the object on the same side as that speaker. The authors suggest that the similar performance during the novel context test provides evidence against this possibility. However, performance during the novel context test could be driven by the formation of an object-label mapping during the endorsement test, which in turn could be based on an attentional bias.*

While we agree that this is an interesting possibility, we do not think it is plausible as an explanation of the pattern of results we have found for a number of reasons. A) More than half the children who selected the object initially labelled by the speaker who turned out to be reliable explicitly justified their object selection in reference to speaker reliability. B) Almost all children passed the trait questions about both speaker's reliability. C) Children did not attend more to the reliable than to the unreliable speaker during the accuracy exposure. Finally, D) We did not provide any feedback after the endorsement test.

- *The authors further argue that their looking time data provides evidence against an attentional bias account, as there was no relation between attention during the accuracy*

exposure and performance in the endorsement test. This again prompts my question about whether the attentional bias accounts specifically predict overt attention to the more reliable speaker? Possibly, a more robust test would be to remove the endorsement test and move straight to the novel context test – which cannot be contaminated by attention to the reliable speaker.

See response to the first comment of reviewer 2.

We agree with your suggestion that a more robust test might have been to skip the endorsement test and move straight to the novel context test. However, for the reasons described above (see esp. response to the first comment of reviewer 2) we do not think this is crucial for the general interpretation of our findings.

MINOR

- *The authors argue that their findings with ‘even younger children’ are consistent with the literature on trait reasoning in children. It would be helpful to know the ages of children in these previous studies.*

Thank you for pointing this out, we added children’s age in the study we are referring to with this statement (p. 16, paragraph 3 and 4).

- *Why is the Luchkina et al (2018) study not mentioned in the introduction? It appears relevant to motivating the present study.*

We now mention the Luchkina et al. (2020) study in the introduction (p. 5, paragraph 2).

Additional changes and clarifications

In addition to the changes suggested by the reviewers above, we incorporated the following minor changes to clarify and improve the manuscript:

- Added additional relevant reference to Heyes (2016, *Dev. Sci.*) in relevant in-text citation (p. 4, paragraph 2) as well as the reference list
- Rephrased report on similar studies concerning early belief revision (p. 5, paragraph 2)
- Added length of the experimental session (p. 11, paragraph 4)

- Corrected linear weighted kappa for children's consistency in object choice between the two retrospective tests (p. 12, paragraph 2)
- Fixed reference list: Luchkina et al. (2020), Kimura & Gopnik (2018)
- Changed figure 2. It now displays 3 groups of children: the ones that chose the object labelled by the reliable speaker, that chose the object labelled by the unreliable speaker and that made no choice
- Table 1: Added number of children justifying their object choice with vs without reference to speaker reliability, divided by prospective test performance (first table on p. 22, last table caption on p. 26f.)
- Re-named variables in data as well as improved clarity of analysis code (on OSF)

Kind regards,

Friederike Schütte, Nivedita Mani, & Tanya Behne